# Influence of the Solar Spectra Models on PACO Atmospheric Correction

Raquel De Los Reyes [1,*], Rudolf Richter [1], Martin Bachmann [2], Kevin Alonso [1], Bringfried Pflug [3], Bruno Lafrance [4] and Peter Reinartz [1]

1 Photogrammetry and Image Analysis, Remote Sensing Technology Institute, Earth Observation Center, German Aerospace Center (DLR), Oberpfaffenhofen, 82234 Wessling, Germany; rudolf.richter@dlr.de (R.R.); kevin.alonsogonzalez@dlr.de (K.A.); peter.reinartz@dlr.de (P.R.)
2 Remote Sensing Data Center, Earth Observation Center, German Aerospace Center (DLR), Oberpfaffenhofen, 82234 Wessling, Germany; martin.bachmann@dlr.de
3 German Aerospece Center (DLR), Earth Observation Center, Remote Sensing Technology Institute, Photogrammetry and Image Analysis, 12489 Berlin, Germany; Bringfried.Pflug@dlr.de
4 CS Group France, 31506 Toulouse, France; bruno.lafrance@csgroup.eu
* Correspondence: raquel.delosreyes@dlr.de; Tel.: +49-81-532831-44

**Abstract:** The solar irradiance is the source of energy used by passive optical remote sensing to measure the ground reflectance and, from there, derive the ground properties. Therefore, the precise knowledge of the incoming solar irradiance is fundamental for the atmospheric correction (AC) algorithms. These algorithms use the simulation results of a model of the interactions of the atmosphere with the incoming solar irradiance to determine the atmospheric contribution of the remote sensing observations. This study presents the differences in the atmospherically corrected ground reflectance of multi- and hyper-spectral sensors assuming three different solar models: Thuillier 2003, Fontenla 2011 and TSIS-1 HRS. The results show no difference when the solar irradiance model is preserved through the full processing chain. The differences appear when the solar irradiance model used in the atmospheric correction changes, and this difference is larger between some irradiance models (e.g., TSIS and Thuillier 2003) than for others (e.g., Fontenla 2011 and TSIS).

**Keywords:** spectral solar irradiance models; atmospheric correction; surface reflectance retrieval; Thuillier 2003; Fontenla 2011; TSIS-1 HRS; Sentinel-2; DESIS





## 1. Introduction

Passive optical remote sensing, by spaceborne or airborne instruments, measures the spectral radiance at the entrance optics of the sensor. This spectral radiance is typically measured between 400–2500 nm. It results from the scattering and absorption of the solar radiation in the atmosphere and Earth ground reflected radiation.

Over the years, many satellite missions have been dedicated to the measurement of the spectral solar irradiance at different spectral resolutions and wavelength ranges which, combined also with modeled data, provide more and more accurate solar irradiance spectra [1–5]. The most recent measurements are provided by the Total and Spectral Solar Irradiance Sensor-1 (TSIS-1) Hybrid Solar Reference Spectrum (HSRS) [6] (called the TSIS 2021 solar model, here simply named TSIS).

Over the reflective range (400–2500 nm), the main energy source for the remote sensing technique is the sun, and therefore a solar irradiance spectrum representative of the illumination conditions is necessary to obtain accurate results for the retrieval of atmospheric parameters, i.e., aerosol optical thickness (AOT) at 550 nm, water vapor column, in order to calculate the ground reflectance. These are the so called L2A products in DESIS and Sentinel-2 data processing chains. L2A products of these two sensors contain the ortho-rectified ground reflectance and parameters that characterize the atmospheric conditions during the acquisitions.

However, we will demonstrate in this study that the most important condition is that the same solar model should be used in the simulation of the atmospheric processes that will create our model for the correction of the atmosphere contribution (L2A processor) of the at-sensor spectral radiance. This ortho-rectified at-sensor radiance (or reflectance, in case of Sentinel-2) is the so called L1C products.

While reference [7] investigates the impact of different solar models on the inter-calibration of satellite instruments, this paper deals with the influence of solar models on the L2A processors. The contribution evaluates the differences in the L2A products with a template scene of Sentinel-2 (multispectral data) and DESIS (hyperspectral data). The comparison is performed with the PACO software [8] using the Thuillier [2], Fontenla 2011 [4] and TSIS solar irradiance models.

The next section (Section 2) describes the solar irradiance models considered in this study and their differences. Section 6 will show the results of the atmospheric correction analysis, i.e., the L2A product, for a sample of a Sentinel-2 and a DESIS scene [9] for both solar models.

The last section will discuss and summarize the similarity of the BOA reflectance when the atmospheric correction is perform with a solar irradiance model, consistent to the rest of the remote sensing data processing. It will also show the level of discrepancy when using different solar irradiance models in the processing chain.

## 2. Solar Irradiance Models for Atmospheric Correction

As the sun is the source of energy for passive remote sensing instruments, a solar irradiance spectrum representative on the acquisition conditions is necessary to obtain accurate results of the ground surface reflectance (Figure 1) retrieved with the atmospheric correction of the data.

The retrieval includes the calculation of the Aerosol Optical Thickness at 550 nm (AOT), atmospheric water vapor column (WV), and in particular the Bottom-Of-Atmosphere (BOA) surface reflectance ($\rho$), used in further stages of Earth Observation research to derive other quantities, such as, for example, the fraction of absorbed photo-synthetically active radiation FAPAR ([10]). Figure 1 presents a schematic sketch of the radiance calibration components, including the solar radiation reflected on the ground target pixel, the intrinsic atmospheric radiance and the surface-atmosphere interaction of the surroundings (adjacency effects). The measured radiance is related to the recorded Digital Number (DN) by the coefficients $c_0$ (offset) and $c_1$ (physical gain depending on the absolute calibration).

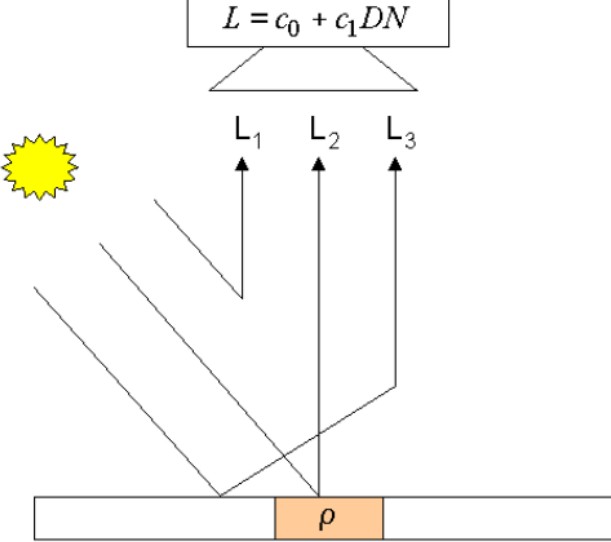

**Figure 1.** Schema of the solar radiative components ($L_i$) in a Lambertian flat-terrain.

Similarly, also the subsequent correction of atmospheric effects (Section 4) is based on the usage of a specific solar irradiance model. The atmospheric correction in software codes such as PACO, a Python version of ATCOR [11], uses LUTs with radiative transfer (RT) functions of atmospheric simulations (Section 4.1). These simulations are performed assuming a certain solar irradiance model.

Therefore, the atmospheric correction should use the same solar model as used for the estimation of the L1 TOA (Top-Of-Atmosphere) radiance (see Section 3).

*Solar Irradiance Models*

In software packages such as PACO and ATCOR, the radiative properties of the atmosphere are simulated in a first stage using radiative transfer (RT) programs such as MODTRAN [12]. For these two AC programs, the simulated RT functions are stored in binned Look-Up-Tables (LUTs) with a spectral sampling distance (SSD) of 0.4 nm. These high resolution LUTs are so called monochromatic and they will be convolved with the spectral response function (SRF) of each channel of the supported sensor.

As the current standard in PACO and ATCOR, the monochromatic LUTs are generated using the high-resolution solar spectral irradiance of Fontenla 2011 [4] (sun medium2 activity, lsunfl = 'A' option of MODTRAN). But other solar models are also supported during the convolution of the monochromatic LUTs to the sensor LUTs according to the remote sensing mission.

This study will compare the results of three solar models:

1. Fontenla 2011 [4] (current default), used in the DESIS L2A processor'
2. Thuillier2003 [2], used by Sentinel-2'
3. TSIS2021 [6], recently recommended by CEOS (Committee on Earth Observation Satellites).

The radiative transfer functions (path radiance, direct and diffuse solar flux, direct and diffuse transmittance, etc) are converted into sensor LUTs depending on the channel spectral response functions.

Based on the latest studies at the time of the mission design, different missions might use different solar irradiance models in their processing chains. For example, Sentinel-2 uses Thuillier2003 while DESIS preferred Fontenla-2011.

The spectral differences between these three solar irradiance models are shown in Figure 2. The three models evaluated in this study are displayed in the upper plot. However, the spectral differences are more visible in the $\delta E_0$ (relative difference, in %) between two of the models (Thuillier2003 and TSIS) with respect to the default one (Fontenla 2011, bottom plot). The spectral differences between models will change the energetic reference at the different wavelengths when compared to the remote sensing measurements. In addition, this difference with the simulations will affect the final calculated ground spectral results.

The major differences between the three models, at a spectral sampling distance of SSD = 0.4 nm and full width at half maximum FWHM = 0.4 nm, happen in the blue wavelengths ($\lambda < 480$ nm). At those wavelengths the difference with TSIS is up to 6% (compared with 10% of Thuillier). For the rest of the wavelengths, Fontenla 2011 and TSIS (orange curve in the lower plot) agree within 2% and below 1% up to NIR wavelengths. In case of Thuillier and TSIS the difference is around 4% in the green and in the SWIR wavelengths. Recently, discussions within GSICS (Global Space-based Inter-Calibration System) and CEOS have resulted in the recommendation of the TSIS solar irradiance spectrum, which seems to have a higher accuracy. This has resulted in a re-evaluation of the solar models used for remote sensing purposes and their effect on the atmospheric correction.

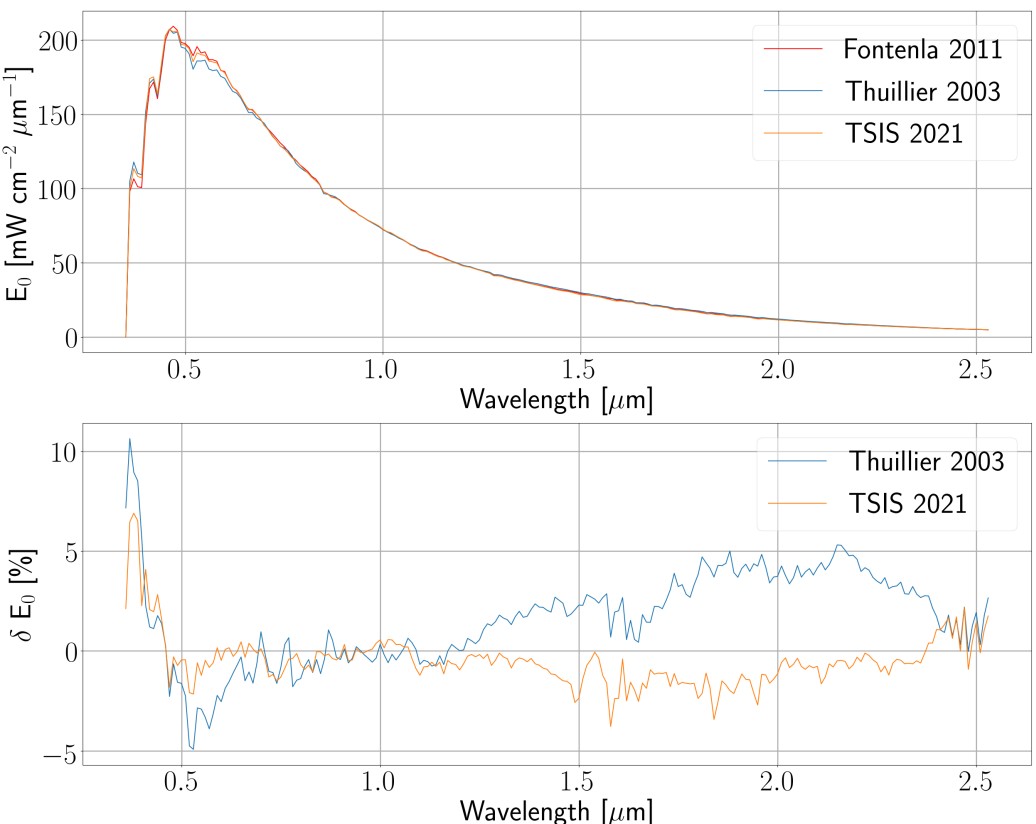

**Figure 2. Top**: Thuillier (blue), Fontenla (red) and TSIS (orange) solar spectra with a SSD = 10 nm and convolved to a Gaussian SRF with FWHM = 10 nm. **Bottom**: Relative solar irradiance differences (in %) of Thuillier (blue) and TSIS (orange) with respect to Fontenla.

## 3. Data for the Atmospheric Correction: L1C Products

For many sensors (e.g., MODIS, Landsat OLI, Sentinel-2 MSI, EnMAP), the absolute radiometric calibration is carried out with solar diffuser panels using the sun as the primary reference. Within the calibration workflow, for each spectral band, the absolute radiometric calibration coefficient is then related to the assumed solar irradiance based on a solar model. Therefore, the choice of the solar model affects all calibrated products of a mission in radiance units, and it has to be considered within further processing and validation workflows [13]. These calibrated products (so called L1 products) contain remote sensing information at the top of the atmosphere. For missions, such as Sentinel-2 and DESIS, they are divided into L1B (sensor geometry) and L1C products. These last ones are the ortho-rectified L1B products.

In particular, the L1 processing chain of Sentinel-2 leads to L1 TOA reflectance products (Section 3) which are solar model independent. The conversion from radiance to reflectance values introduces a normalization of the radiance values by the solar irradiance spectra. Therefore, when the user works with TOA radiance values, the common solution is to use the solar irradiance values provided with auxiliary data to convert the provided TOA reflectance values to radiance.

The input to the PACO/ATCOR programs is a spectral image cube containing the TOA radiance (L1 = Level 1), either in sensor geometry (L1B product) or ortho-rectified (L1C product), and in units of ($mW \cdot cm^{-2} \cdot sr^{-1} \cdot \mu m^{-1}$). If scaling factors are used, then the input radiance is calculated band-wise as $L = c_0 + c_1 DN$ (offset $c_0$, gain $c_1$, Digital Number $DN$). For this study, we evaluate ortho-rectified Sentinel-2 data (in TOA reflectance) and DESIS data (in TOA radiance).

The L1C products of Sentinel-2 and DESIS are different and they are summarized below.

### Sentinel-2

The Sentinel-2 mission uses the Thuillier 2003 solar model for the data processing.

The ESA L1C products from Sentinel-2 are expressed in TOA reflectance ($\rho^*$) (Equation (1)) based on the Thuillier irradiance spectrum, but multiplied with a scale factor of 10,000. The relation between radiance (L) and reflectance ($\rho^*$) values is given by Equation (1).

$$\rho_k^*(i,j) = \frac{\pi \times L_k(i,j)}{E_0(k) \times d(t) \times \cos(\theta)} \qquad (1)$$

where $L_k$ (in mW·cm$^{-2}$·sr$^{-1}$ · μm$^{-1}$) and $E_0$ (in mW·cm$^{-2}$ · μm$^{-1}$) are the TOA radiance and the equivalent extraterrestrial solar irradiance spectrum per each Sentinel-2 band (k), respectively. $d(t)$ is the sun–Earth distance orbital variation and $\theta_s$ is the solar zenith angle. A change of the solar model $E_0$ (in Equation (1)) induces a change of the value of TOA radiance conversion ($L_k$) but it does not affect $\rho^*$, leaving this last one independent of the solar model spectra.

For Sentinel-2 data, the TOA reflectance is converted into TOA radiance during the PACO/ATCOR L2A processing applying the mission-dependent solar irradiance model or any other model we want to study (see Sections 6.1 and 6.2).

Sentinel-2 spectral data consists of 13 bands at different wavelengths and three different spatial resolutions (10 m, 20 m, 60 m) [14]. In order to make a complete spectral analysis of Sentinel-2 L2A results, for this study, we have merged the spectra of all the Sentinel-2 bands in a merged simulated sensor. The spectral bands of the 10 m and 60 m data cubes have been interpolated spatially and merged to a final 20 m resolution data cube with 13 spectral bands.

### DESIS

On board of the ISS, the DLR Earth Sensing Imaging Spectrometer (DESIS) observes the Earth in the spectral range between 400–1000 nm with a spectral sampling distance (SSD) of 2.55 nm and an FWHM of about 3.5 nm. The observation swath of 30 km is covered with a ground sampling distance (GSD) of 30 m.

The DESIS L1C products are given in radiance with the unit (mW·cm$^{-2}$·sr$^{-1}$ · μm$^{-1}$), and the corresponding LUTs for atmospheric correction are based on Fontenla-2011 [4].

Since the DESIS L1C products are not solar model independent, an atmospheric correction processor based on radiances would only need to ensure to process the DESIS data using the same solar model as the mission.

Therefore, in this study, we will illustrate the effects on the DESIS L2A results (see Section 6.1) when using the DESIS L1C radiance products and another solar model (TSIS) (see Section 6.2).

## 4. Atmospheric Correction Algorithms: PACO/ATCOR

The spectral differences of both solar models (Fontenla/Thuillier) will not only be translated into differences in the radiative transfer LUTs, but also might change the L2A BOA reflectance results.

The atmospheric correction is performed using a set of bands at different wavelengths to characterize different atmospheric parameters (Aerosol Optical Thickness AOT and water vapor WV) as well as classify the nature of certain pixels to be used/excluded in such algorithms (e.g., Dark Dense Vegetation (DDV), excluded clouds, shadows, etc.).

As it is seen in Figure 2, the critical bands that might affect the L2A products are the blue, green and the SWIR bands (1.6–2.3 μm), for which the choice of the solar irradiance model has an influence of 2–4%. These bands are used in our L2A processor in different algorithms and their difference must be studied to understand the possible effects in the L2A results.

Processing algorithms based on the TOA apparent reflectance will not be significantly affected in the case of Sentinel-2 since the L1C is already given in TOA reflectance. Only

those algorithms that use the BOA reflectance (flat-terrain or Lambertian) or TOA radiance might be affected, since also the RT functions are affected (see as example Figure 3).

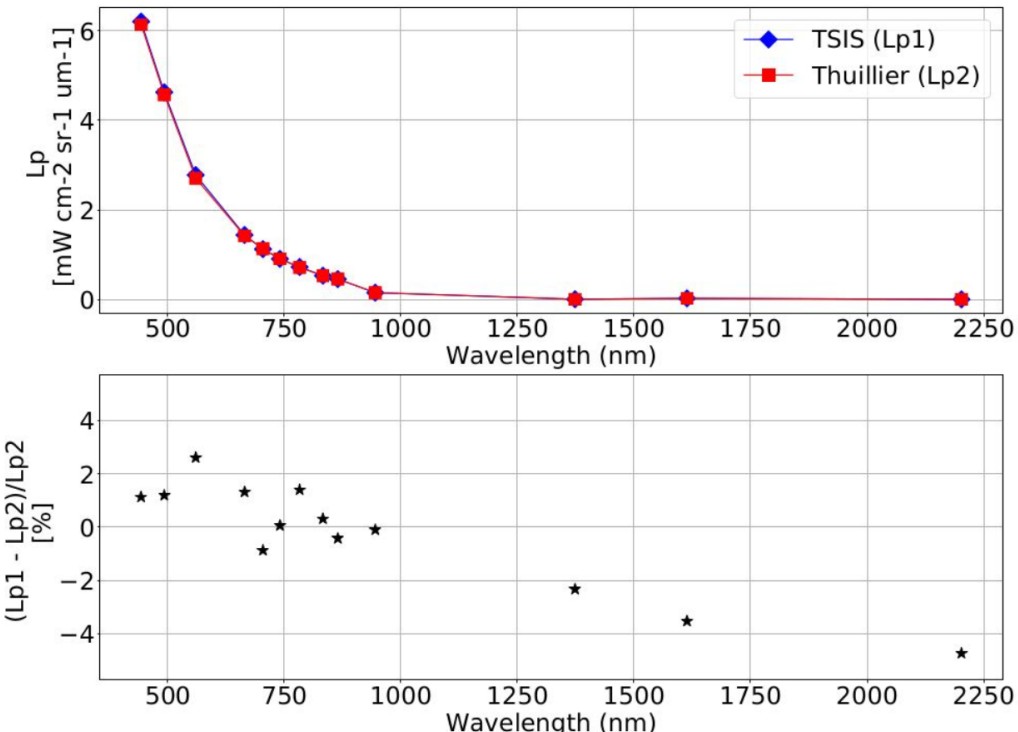

**Figure 3.** **Top**: Atmospheric path radiance for Sentinel-2 A bands using Thuillier 2003 (red line) and TSIS (blue line) solar models. **Bottom**: Relative difference (black stars) between the convolved atmospheric path radiances.

Those processes are listed here and they will be examined more in detail in this study:

- **Pre-classification**: haze and water pixel masks. The haze mask could also be affected since it uses a relationship between the first two bands (in radiance DN units). In case of the water mask, the implemented threshold in BOA reflectance at 1.6 μm to mask water pixels might also create differences in the pre-classification results between both solar irradiance models.
- **Dark Dense Vegetation (DDV) pre-classification**: the selection of DDV pixels uses the BOA flat-terrain reflectance in the SWIR (2.1 or 2.2 μm) and NIR (860 nm) bands, and the atmospheric path radiance ($L_p$) for the DDV NIR algorithms.
- **AOT estimation**: the estimation of the AOT assumes an invariant ratio between red and the SWIR/NIR [15,16] bands for the previously classified DDV pixels. Any variation that alters the scale factor between the red and the NIR/SWIR BOA reflectance would have an impact on the estimation of the AOT.

The significant differences in one or more intermediate steps of the previous processes might increase the uncertainty of the final product: the Bottom-of-Atmosphere surface reflectance.

However, in a flat-terrain BOA reflectance calculation the ratio between the mission and the solar model under study would cancel since the radiative parameters $L_p$ (atmospheric path radiance), $E_{dif}$ and $E_{dir}$ (diffuse and direct irradiance, respectively, illuminating the surface) involved in the BOA reflectance inversion are linearly dependent on the solar irradiance and therefore multiplied by the same ratio.

### 4.1. Radiative Transfer Functions: Sentinel-2

Figure 4 shows the CEOS-recommended solar model TSIS versus the current one (Thuillier 2003) for the Sentinel-2 bands. The relative difference per Sentinel-2 band between

both models as a function of the wavelength is displayed in the plot below. The convolved spectra reproduced the results of Figure 2, with a difference of 2% at green wavelengths and ∼4% at SWIR wavelengths.

The radiative LUTs ($L_p$, $E_{dif}$ and $E_{dir}$) corresponding to Sentinel-2 bands show the same order of magnitude (2–4%) in the relative difference for the already identified critical wavelengths (bottom plot in Figures 3 and 4). Figure 3 shows as an example, the atmospheric path radiance for a scene with AOT ∼ 0.3 at sea level (visibility of 23 km), ground altitude at sea level, sun zenith angle (SZA) = 30°, relative sun-viewing azimuth angle (SAA) = 60° and off-nadir viewing angle = 10°.

The direct ($\tau_{dir}$), diffuse ($\tau_t extdif$) and sun-to-ground ($\tau_{sun}$) atmospheric transmittance, and the spherical albedo are invariant with respect to the solar model.

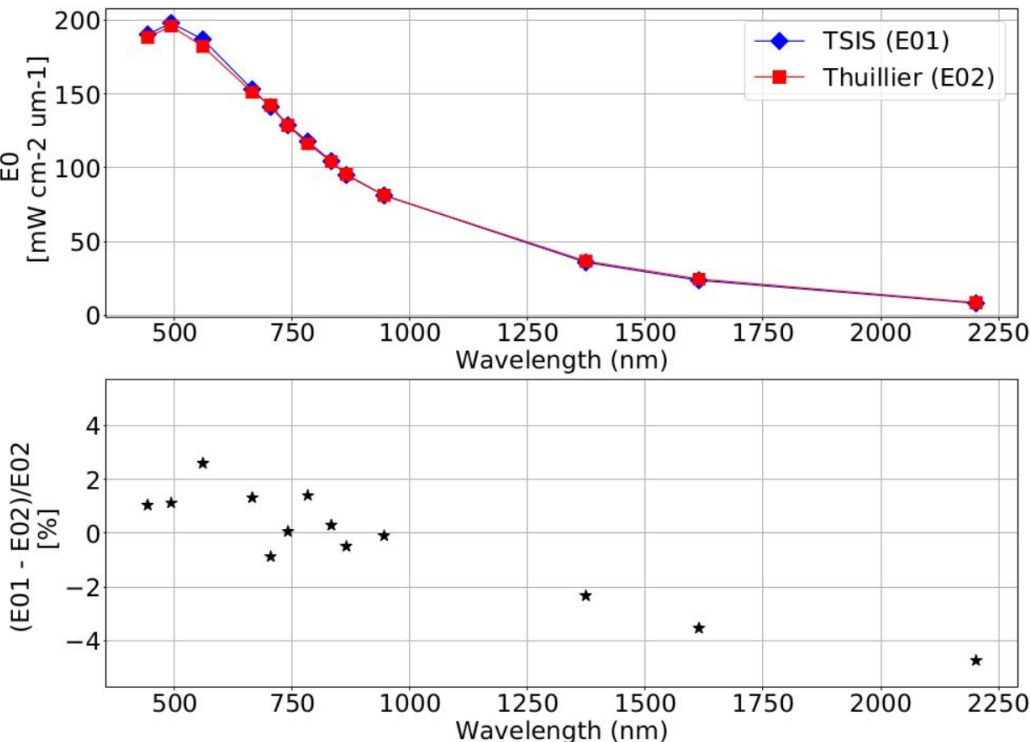

**Figure 4. Top**: Solar irradiance spectra for Sentinel-2A bands using Thuillier 2003 (red line) and TSIS (blue line) solar models. **Bottom**: Relative difference (black stars) between the convolved solar models.

## 5. Atmospheric Correction Scenarios under Study

The important point is that for each sensor, the processing chain uses the same solar irradiance ($E_0$ in Equation (1)), both for the estimation of the TOA radiance and for the process of atmospheric correction or for the retrieval of geophysical parameters. Otherwise, these inconsistencies in the solar models lead to incorrect radiance values at the TOA level [13].

To prove this statement, this study shows the consequences in the final L2A products assuming two scenarios. They correspond to the consistent and the inconsistent radiance handling of L2A processor within the mission processing chain. Therefore, in both scenarios the atmospheric correction will be conducted using the TSIS and the corresponding mission solar model. The only difference is the assumed input:

- **Mission L1C radiances**: the original L1C radiances pertaining to the mission solar model (M1) are the input to the atmospheric correction, and its LUTs are produced with solar model M2. *Inconsistent* scenario.
- **Updated L1C radiances**: updated L1C radiances for irradiance model M1 or M2 consistent with the LUTs based also on the same model M1 or M2. *Consistent* scenario.

Therefore, in the first scenario there will be an inconsistency in the processing since the radiative scenario is not consistent through the processing chain. The second scenario might imply an update of the L1C radiance products but it will preserve the consistency between the L1C radiance and the RT functions in the LUTs.

Sentinel-2 data will be used in both scenarios to compare the differences in the L2A products. The DESIS products will be used only in the *inconsistent* or *Mission L1C radiances* scenario (Section 6.2), since the *consistent* scenario will be proven in detail in the section before *Updated L1C radiances* (Section 6.1).

## 6. Results

For this study, we choose Sentinel-2 and DESIS data acquisitions over the RadCalNet [17] site La Crau in order not only to inter-compare between the final remote sensing surface reflectance products but also use in situ measurements (RadCalNet or RCN) as a reference.

The surface reflectance uncertainty of PACO for multi-spectral missions is estimated to be $U_{S2,BOA}[\%/100] \leq 0.05 * BOA_{RCN} + 0.005$ [8]. For hyper-spectral VNIR missions, such as DE-SIS, where the sensor spectral range starts at 400 nm, the uncertainty is $U_{DESIS,BOA}[\%/100] = 0.04 * BOA_{RCN} + 0.011$ [18].

Assuming a coverage factor of k = 1, the uncertainty ratio (*K*) will quantify the deviation of the measurements ($Y_{PACO}$) with respect to independent reference data ($Y_{REF}$) in terms of the expanded uncertainty [19] of both measurements. If the probability of the measurements follow a normal distribution, *K* = 1 would represent a difference of $1\sigma$ difference with respect to the combined uncertainty:

$$K = \frac{Y_{PACO} - Y_{REF}}{\sqrt{U_{PACO}^2 + U_{REF}^2}} \tag{2}$$

where *Y* is replaced by the ground surface reflectance ($\rho$) in this study. Assuming the statistical independence of the remote sensing and in situ measurement, the combined uncertainty is the quadratic sum of the uncertainty in both measurands.

For wavelengths < 440 nm, it has been noticed from hyperspectral sensors that a linear regression of the uncertainty bins is not the best model to use due to the contribution of the AOT uncertainty [18]. This leads easily to *K* > 1 when comparing with RadCalNet in situ measurements. Only for scenes where the difference between the AOT estimation of PACO and AERONET/RadCalNet is really small, the *K* value for $\rho$ should approximate to 0 for $\lambda < 440$ nm.

### 6.1. Updated L1C Radiances Scenario: Sentinel-2

As mentioned in Section 3, for the purpose of the present investigation, the atmospheric correction is performed in two ways using the Sentinel-2 L1C TOA reflectance as input. In this section, we will show the results when preserving the radiative consistency of the L1 and L2 processing chains.

This means the L1C radiance is either Thuillier or TSIS and the same solar model is applied to the LUTs during the atmospheric processing.

### 6.1.1. Data

Both solar models will be applied to the same test scene: a Sentinel-2A acquisition over La Crau RadCalNet site (tile T31TFJ), acquired on 29 July 2019. The scene was acquired under a sun zenith angle (SZA) = 28.8°, sun azimuth angle (SAA) = 105.07° and off-nadir view angle = 7.02°. The elevation model used is the SRTM (Shuttle Radar Topography Mission) with 1 arcsec resolution.

The final L2A surface reflectance is Lambertian, since no BRDF correction is applied.

The atmospheric conditions at the time of the overpass measured by RadCalNet are: $AOT_{RCN} = 0.071 \pm 0.001$ and $WV_{RCN} = 1.23 \pm 0.25$ cm.

Table 1 shows the resulting scene statistics of L2A products: the pre-classification, AOT, WV, surface reflectance of DDV reference pixels, and scene NDVI.

**Table 1.** L2A statistical results (mean and standard deviation) from the Sentinel-2 La Crau scene processing using Thuillier 2003 and TSIS solar models (consistent scenario).

| L2A Stats | Thuillier 2003 | TSIS |
|---|---|---|
| Haze / Water / DDV [%] | 58.7 / 15.4 / 13.1 | 58.7 / 15.4 / 13.1 |
| Refl. Reference Pixels (red) [%] | 3.8 | 3.8 |
| $AOT_{scene}$ (550 nm) | 0.09 | 0.09 |
| $Lp_{scene}/Lp_{MODTRAN}$ | 1.02 | 1.02 |
| WV [cm] | $1.74 \pm 0.26$ | $1.74 \pm 0.26$ |
| NDVI | $0.52 \pm 0.10$ | $0.51 \pm 0.10$ |

These results are discussed in more detail in the next sections.

6.1.2. Pre-Classification

The results from Table 1 show no significant difference in the haze mask (marked in yellow in the middle Figure 5). The relative differences are below 0.05%, possibly due to the small relative differences in the spectra between the 440 nm and 492 nm bands employed for haze detection.

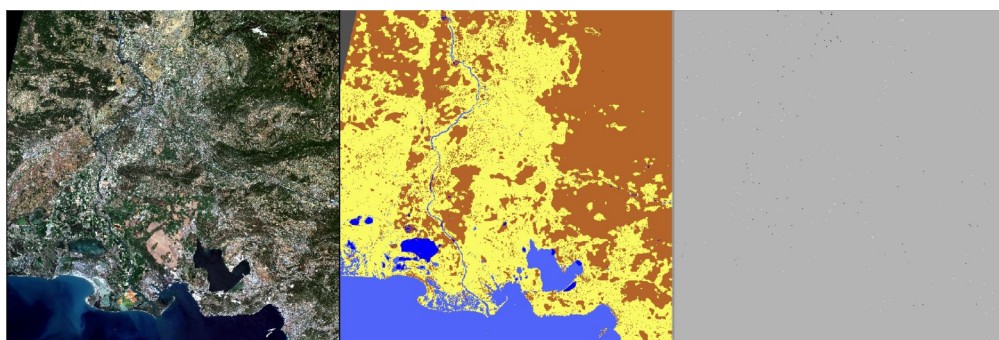

**Figure 5.** L1C RGB (**left**), pre-classification map Thuillier 2003 (**middle**) and pre-classification absolute difference map between TSIS and Thuillier masks product (**right**), where the grey color represents a zero difference. The pre-classification mask tags present in the middle plot are: clear land (brown), clear water (dark blue) and haze over land (yellow) and over water (light blue).

The relative difference of 4% at 1.6 µm between both solar models does not affect significantly the relatively small reflectance values for water pixels. There is a slightly higher percentage classified as water pixels when using the TSIS model ($\sim 10^{-4}$%).

The absolute difference (right plot in Figure 5) shows no systematic differences. For the total number of $5490 \times 5490$ pixels, 99.9% of the pixels show no difference in their mask tag (grey color represents a difference of 0).

DDV pixels are masked (Figure 6) by calculating the surface reflectance at a visibility of 23 km for the 2.2 µm band of Sentinel-2. The DDV reflectance values should be higher than 1% and lower than 5%. Water reflectance values are also small in the SWIR region, but water is excluded using the criterion NDVI > 0.1 and TOA reflectance < 7% in the red band (665 nm).

The distribution of pixels with Thuillier/TSIS differences (right plot in Figure 6) do not show any pattern. 99.97% of the pixels have the same classification flag. 0.02% of the pixels have mismatches between the DDV/no-DDV and the DDV/water pixels tags. The mismatch between no-DDV and water pixels is negligible ($10^{-4}$%).

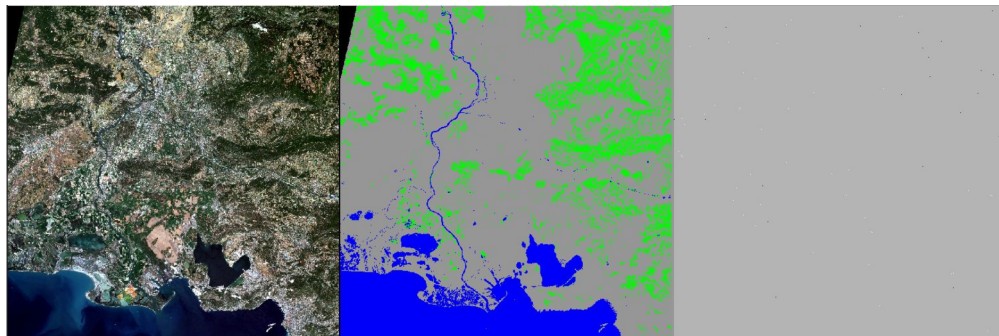

**Figure 6.** L1C RGB (**left**), DDV map Thuillier 2003 (**middle**) and DDV relative difference map between TSIS and Thuillier product (**right**), where the grey color represents a zero difference. The DDV classification masks are: DDV pixels (green), clear water (dark blue) and other (grey).

6.1.3. Aerosol Optical Thickness

The only critical point that could be affected, looking at the relative differences in Figure 3 would be the predefined red-SWIR ratio. The SWIR wavelength is 2.1 μm for Sentinel-2 and the red one is 0.665 μm). Any deviation of the relative reflectance between these two channels might affect the estimated AOT of the DDV algorithm, even if the population of DDV pixels does not change.

Table 1 shows that the reflectance of the DDV reference pixels does not change when switching from Thuillier to TSIS, and therefore also the scene mean DDV reflectance values ("Refl. Reference Pixels (red)") do not change. The AOT is calculated based on the DDV red-SWIR surface reflectance ratio, averaged with a low pass filter of 3 km × 3 km, and applied to the whole scene.

In the right plot of Figure 7, we can distinguish some small differences in the AOT map. This is caused by a small amount of water pixels, added to the water mask when applying the TSIS solar model. These pixels are also present in Figure 5. However, the AOT values estimated from these water pixels are smoothed (3 km × 3 km) together with the one from the DDV pixels in the final AOT map. This low pass filtering process enlarges the single pixel differences to small systematic areas around them.

The AOT map is calculated taking into account the elevation (DEM) of the pixel and the reflectance of the reference pixels, so if there are no changes in the reflectance of the reference pixels between both solar models, then no AOT changes are expected (Table 1).

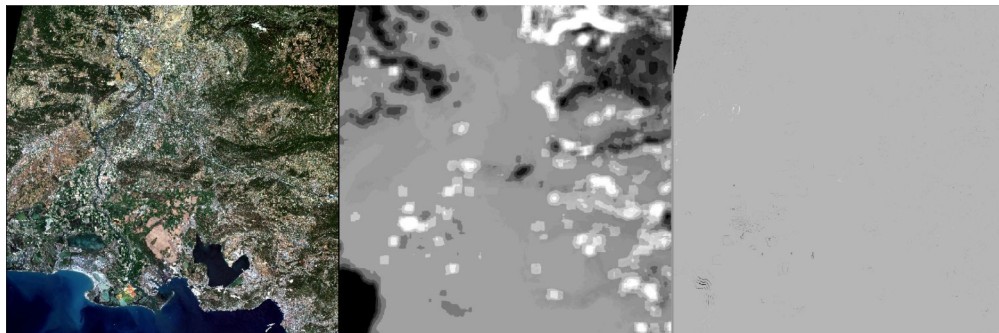

**Figure 7.** L1C RGB (**left**), AOT (@ 550 nm) map Thuillier 2003 (**middle**) and AOT relative difference map between Thuillier and TSIS product (**right**).

6.1.4. Water Vapor (WV)

The small relative changes between Sentinel-2 bands 8a and 9 (865 and 945 nm, respectively) of <1%, might have a small effect in the water vapor retrieval over vegetation areas (see right image on Figure 8). In these areas, the relative difference in the WV retrieval is of ∼0.5% and does not change the difference of the scene water vapor column.

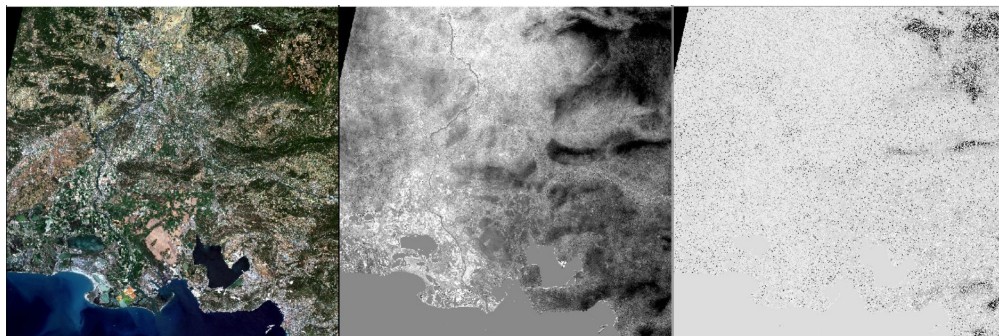

**Figure 8.** L1C RGB (**left**), WV map [in cm] Thuillier 2003 (**middle**) and WV relative difference map between TSIS and Thuillier product (**right**).

This water vapor difference between both solar irradiance models is in any case negligible: smaller than the 1 $\sigma$ uncertainty of the WV retrieval method for multi-spectral sensors such as Sentinel-2: $U_{WV} = 0.02 * WV_{AERO} + 0.13$ [8], according to the uncertainty of the APDA [20] algorithm employed by PACO/ATCOR (WV in cm).

A change of the mean scene WV of <1% would have no effect on the final BOA surface reflectance retrieval, as demonstrated in the next section.

Several scenes were investigated concerning the influence of water vapor on the surface reflectance spectra of soil and vegetation.

The tests demonstrate that for vegetation spectra this translates to an uncertainty typically smaller than 0.003 reflectance units, for agricultural soil it is less than 0.002 reflectance units, except for the water vapor channel (945 nm), where the uncertainties are up to 0.05 units (vegetation) and 0.03 units (soil).

### 6.1.5. BOA Reflectance

Although the RadCalNet site of La Crau is flat, the surface reflectance is calculated with terrain correction [21] using SRTM as DEM reference database with 1 arc-second resolution.

Figure 9 shows the retrieved surface reflectance from RadCalNet data (black "*"s) and the PACO corrected S2 data with the Thuillier (blue "+"s) and TSIS (red dots) solar models.

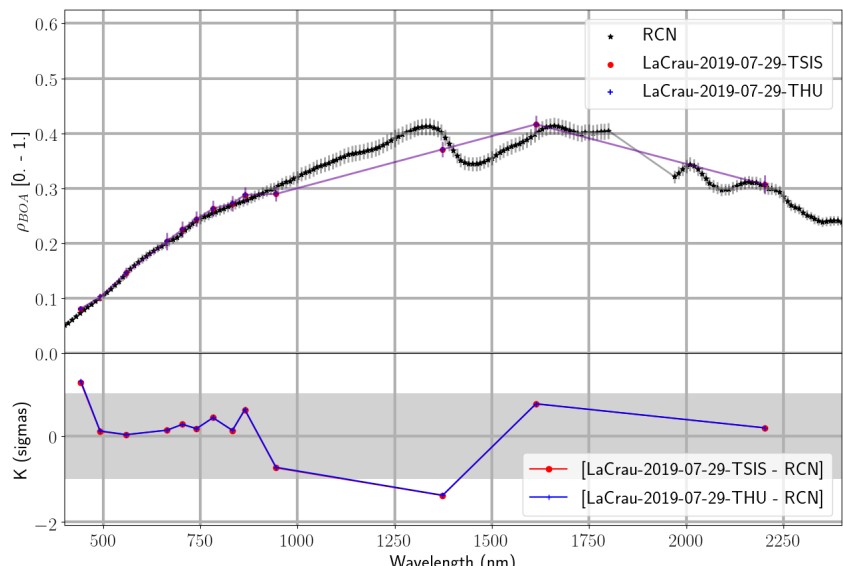

**Figure 9.** Consistent scenario. **Top**: L2A surface reflectance of RadCalNet (RCN) (black), PACO L2A with Thuillier 2003 (blue, "+") and TSIS (red, ".") solar models. **Bottom**: Uncertainty ratio (K) between each of L2A surface reflectance with the previous solar models with respect to RadCalNet in situ reference values. Grey band limits $\pm$ 1 sigma region.

As expected from our previous results, the differences are negligible when using one or the other solar model.

The uncertainty ratio (K) includes not only the uncertainties from the RadCalNet ROI but also the uncertainties from the PACO BOA retrieval [8] (Equation (2)). For the BOA calculation, REF corresponds to RadCalNet.

Both solar models seem to deviate from RadCalNet measurements in the coastal band (~440 nm). This is not due to the solar models but to the underestimation of the uncertainty of the BOA retrieval (by fitting the uncertainty to a power law including the blue wavelengths).

The AOT difference between the PACO ($AOT_{PACO} = 0.09$) and RadCalNet ($AOT_{RCN} = 0.07$) measurements is 0.02. In any case, this is smaller than the typical PACO AOT retrieval uncertainties [8].

These results prove that the estimate of the atmospheric parameters and the surface reflectance are very weakly dependent on the solar model used if the same solar model is used both to estimate the TOA radiance and in the atmospheric simulations (so called *consistent* scenario).

In the next section, we will break this consistency and analyze the influence on the L2A products.

## 6.2. Mission L1C Radiances: DESIS and Sentinel-2

The part of this study consists in the comparison of the L2A products using different solar models for atmospheric correction than the ones used for the radiance estimation of each mission: Thuillier for Sentinel-2 and Fontenla for DESIS. In this study, we compare the mission results with the atmospheric correction using the CEOS recommended solar model TSIS.

We illustrate in the next section the differences in the L2A products if the solar model consistency is not maintained between the measured radiance and the atmospheric correction processing. The exercise is conducted for a VNIR (DESIS) and a VNIR-SWIR (Sentinel-2) sensor.

### 6.2.1. Data

The test acquisitions correspond to a DESIS and Sentinel-2 overpass over the RadCalNet site of La Crau [17] on 15 August 2020 (see Figure 10). A different dataset from the one in the previous section was chosen in order to increase the acquisition scenes analyzed in this study.

Table 2 summarizes the overpass conditions of both sensors over the RadCalNet site and some differences between the sensors. These differences are for example the nominal solar model used for the radiance estimation and the spatial resolution of both sensors. The spatial resolution of Sentinel-2 specified in Table 2 corresponds to the merged cube (see Section 3).

**Table 2.** Summary of the acquisition conditions of the DESIS and Sentinel-2 overpass over La Crau RadCalNet site on 15 August 2020. "#" denotes the total number of pixels.

| L1C Stats | DESIS | Sentinel-2 |
|---|---|---|
| Time [UTC] | 08:54:55 | 10:40:31 |
| Sun zenith, azimuth angle [°] | (47.16, 115.44) | (32.21, 153.98) |
| Sensor off-nadir/azimuth angle [°] | (6.8, 154.13) | (8.1, −72.93) |
| Solar model (radiances) | Fontenla 2011 | Thuillier 2003 |
| Pixel size [m] | 30 | 20 |
| Scene pixels [#] | 1,099,552 | 24,902,219 |

As in the previous exercise, the DEM is SRTM (1″ resolution), but the atmospheric correction is performed without terrain correction for DESIS because less than 1% of the pixels have slopes > 6°.

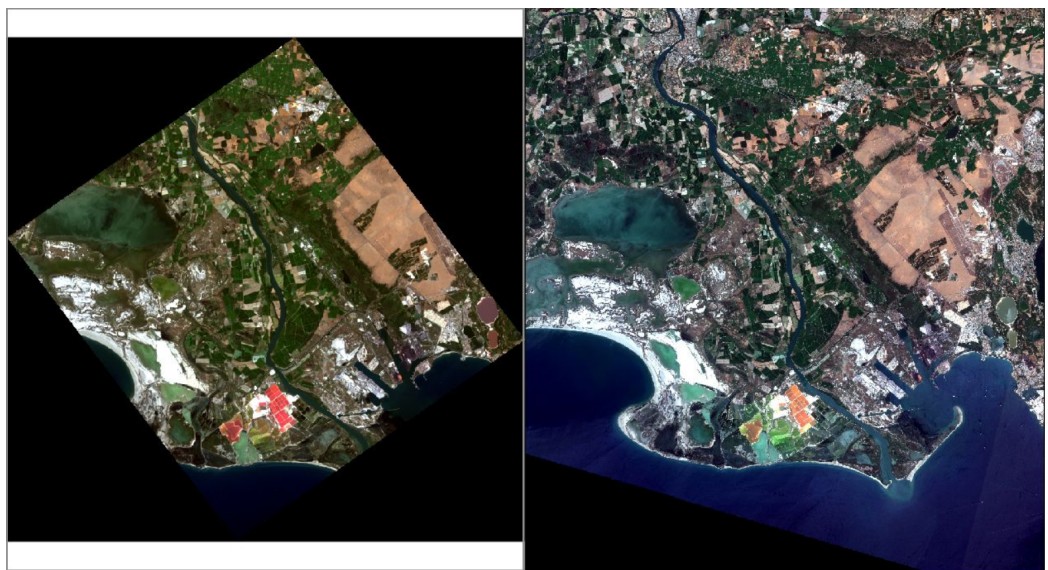

**Figure 10.** L1C RGB of DESIS (**left**) and Sentinel-2 (**right**), acquisitions on 15 August 2020 over La Crau RadCalNet site.

Table 3 summarizes the L2A statistical results for both sensors (DESIS and Sentinel-2) using their mission solar model (Fontenla 2011 and Thuillier 2003, respectively) and for atmospheric correction the TSIS 2021 solar model recommended by CEOS. The results discussed in the next sections illustrate the level of inconsistency in the L2A products as a result of not following the consistent mission solar model. Both solar models yield different levels of disagreement with respect to the CEOS recommendation (TSIS) (see Figure 2).

The comparison between the different results must be performed quantitatively between solar models for the same sensor, but it can be conducted only qualitatively between the sensors. The reason is the different wavelength range covered by both sensors. The Sentinel-2 bands in the SWIR wavelength range enable more precise results in some algorithms such as the AOT retrieval and for some masks (e.g., snow/cloud detection, etc). This will be discussed in the corresponding section.

**Table 3.** Comparison of PACO L2A statistics for DESIS and Sentinel-2 atmospheric correction with their mission solar model (Fontenla 2011 and Thuillier 2003) and TSIS 2021 (Inconsistent scenario).

| L2A Stats | DESIS | | Sentinel-2 | |
|---|---|---|---|---|
| | **Fontenla 2011** | **TSIS** | **Thuillier 2003** | **TSIS** |
| Snow [%] | 0.3 | 0.3 | 0.07 | 0.06 |
| Cloud [%] | 0.8 | 0.8 | 0.1 | 0.1 |
| Haze [%] | 15.3 | 15.3 | 16.6 | 16.6 |
| Water [%] | 27.7 | 27.9 | 12.6 | 12.6 |
| DDV [%] | 3.6 | 3.7 | 3.3 | 2.3 |
| Refl. Reference Pixels (red) [%] | 2.05 | 2.05 | 2.2 | 2.2 |
| $AOT_{scene}$ (550 nm) | $0.095 \pm 0.03$ | $0.095 \pm 0.03$ | $0.09 \pm 0.02$ | $0.09 \pm 0.02$ |
| $Lp_{scene}/Lp_{MODTRAN}$ | 1.11 | 1.14 | 1.05 | 1.04 |
| WV [cm] | $2.23 \pm 0.69$ | $2.17 \pm 0.67$ | $2.16 \pm 0.50$ | $2.17 \pm 0.50$ |

### 6.2.2. Pre-Classification

In order to compare the pre-classification statistics one has to consider the total amount of pixels covered by the detector swath (scene pixels in Table 2) for each sensor.

So, an atmospheric phenomenon such as haze could have closer results between sensors if the phenomenon is homogeneously distributed for the full sensor tiles in both acquisitions. Other phenomena related with the ground characteristics can not be compared since the ground area covered by both sensors is slightly different.

In the pre-classification, a large difference between the sensors in the amount of snow and cloud pixels already reveals the limitations in the consistent classification when SWIR bands are missing.

Some of those pixels are actually bright sand areas, that are confused as snow for the masking algorithms in sensors with SWIR bands (Sentinel-2) and like clouds for sensors without SWIR bands such as DESIS.

The haze mask is calculated for both sensors at the 0.5 $\sigma$ level of the combined haze and blue bands. However, the haze band for DESIS is at 430 nm, while for Sentinel-2 the analysis only has the coastal band (B1) at 443 nm. This could explain the slight difference in the haze mask even if the haze distribution would be homogeneous in both cases.

### Sentinel-2

Comparing the pre-classification results in the last two columns of Tables 1 and 3, we see the inconsistencies when L2A products are not processed with the correct solar irradiance model pertaining to each mission.

The last two columns of Table 3 show nearly no differences between the masks statistics. In most of the masks the differences are of the same order of magnitude in Section 6.1.2. Even considering that in this part of the study there are already TOA reflectance differences. The small differences might be due to the small differences in the solar model irradiance. Although the difference in the SWIR bands is up to 6%, the solar irradiance at those wavelengths are typically small (see Figure 2).

### DESIS

The closer agreement between TSIS and Fontenla 2011 solar models (Figure 2) causes less inconsistency results.

### 6.2.3. DDV and AOT

The AOT results for both solar models and sensors are influenced by two factors: the population of DDV pixels and the calculation of surface reflectance of the reference pixels (DDV) from the SWIR [15] and VNIR [16] bands for Sentinel-2 and DESIS, respectively.

For both sensors the AOT information is extracted from the reflectance at the red band, derived by a 0.5 and 0.1 factor from the SWIR (2.2 µm) [15] and VNIR bands (865 nm) [16], respectively (Equation (3)).

$$\rho_{red}(S2) = 0.5 \cdot \rho_{2.2\mu m}; \qquad \rho_{red}(DESIS) = 0.1 \cdot \rho_{865nm} \tag{3}$$

Since the AOT estimation algorithms for VNIR (DESIS) and SWIR (S2) are different, the AOT estimation from DESIS and Sentinel-2 in Table 3 can only be compared qualitatively.

Nevertheless, the results for both sensors (Table 3) are within the PACO design requirements [8] when compared with the AOT of the RadCalNet site: $AOT_{RCN} = 0.076 \pm 0.001$.

### Sentinel-2

As we saw previously, the differences are still negligible for most of the masks, except for the DDV pixels. The difference of 3–4% in the TSIS solar model at SWIR wavelength (2.2 µm) creates an inconsistency in the amount of DDV pixels, and therefore in the AOT.

However, the difference between the AOT values ($AOT_{TSIS} = 0.087$) and the Thuillier one ($AOT_{Thuillier} = 0.093$) is negligible. The difference is significantly smaller than the typical AOT uncertainty in this AOT range ($U_{AOT} \sim 0.06$) [8]. Therefore, in the statistics in Table 3 we round the results to the significant decimal, giving no numerical difference.

### DESIS

The small differences between Fontenla 2011 and TSIS create consistent results in the population of DDV pixels. For VNIR sensors, the AOT is estimated from the VNIR bands and the reflectance of the reference pixels in the red band is a constant factor of 0.1 times the reflectance at the VNIR bands (Equation (3)). The difference around 860 nm between

Fontenla and TSIS irradiance spectra (Figure 2) is negligible, and therefore the AOT values are the same.

### 6.2.4. Water Vapor

The wavelength range of both sensors also determines the differences in the water vapor retrieved using the APDA algorithm [20] for both sensors. Sentinel-2 uses the bands 8a (865 nm) and 9 (945 nm) while DESIS, due to some ethaloning effects at those wavelengths [9], uses the water vapor absorption region at 820 nm.

The results between both sensors and between models are quite similar, and smaller than the uncertainty of the WV estimation algorithm in this range: $U_{WV}$ = 0.17 cm [8]. Also the water vapor results are within the uncertainty when compared with the water vapor measured in situ: $WV_{RCN}$ = 2.31 $\pm$ 0.00 cm. As discussed in Section 6.1.4, these differences will not affect the final BOA retrieval.

### 6.2.5. BOA Surface Reflectance

As in Section 6.1.5, the resulting BOA surface reflectance is compared for both sensors with the same La Crau in situ measurements.

Since different algorithm are employed for DESIS and Sentinel-2, a direct comparison of the BOA reflectance results is difficult. Therefore, the results of both sensors are compared with the in situ reference measurements.

Sentinel-2

The result of changing the solar model with respect to the one used for the radiance estimation can be seen comparing Figures 9 and 11.

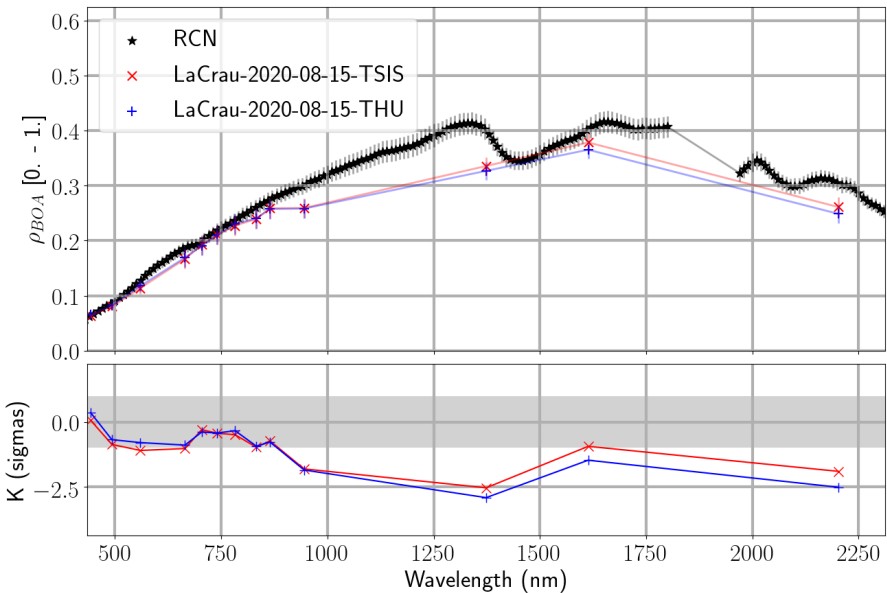

**Figure 11.** Inconsistent scenario for Sentinel-2. Top: L2A surface reflectance of RCN (black crosses), PACO L2A with Thuillier 2003 (blue, "+") and TSIS (red "x") solar models. Bottom: Uncertainty ratio (K) between each of L2A surface reflectance with the previous solar models with respect to RadCalNet (RCN) in situ reference values. Grey band limits $\pm$ 1 sigma region.

In the first one, the absolute calibration was preserved by using the same solar model to convert from TOA reflectance to radiance and the atmospheric correction (LUTs). The L2A surface reflectance was the same using both solar models.

In this last exercise (Figure 11), one can see the increase in the difference in the SWIR bands (1.6 µm and 2.2 µm) with respect to the in situ measurements (RCN La Crau in black

crosses) depending on the solar model used. In addition, some differences are visible in the green bands.

The trend of the deviations follows the trend of the solar irradiance models and their relative differences (Figure 4): positive (TSIS > Thuillier) at green bands and negative (TSIS < Thuillier) in the SWIR bands.

This result shows the inconsistency of using different solar models to convert to radiance and simulations (LUTs) for the atmospheric correction.

DESIS

The small differences between the TSIS and the Fontenla 2011 solar models result in no significant difference between the BOA surface reflectance retrieved by DESIS using either one of the models (see Figure 12).

Both solar models deviate by the same amount from the in situ reference below $\lambda < 500$ nm. However, below these wavelengths the deviation is due to an underestimation of the uncertainty at those wavelengths and the dependency of the latest with the AOT uncertainty. In addition, some spectral details in the Fraunhofer lines in the TSIS solar model are visible for DESIS (FWHM = 3.5 nm)

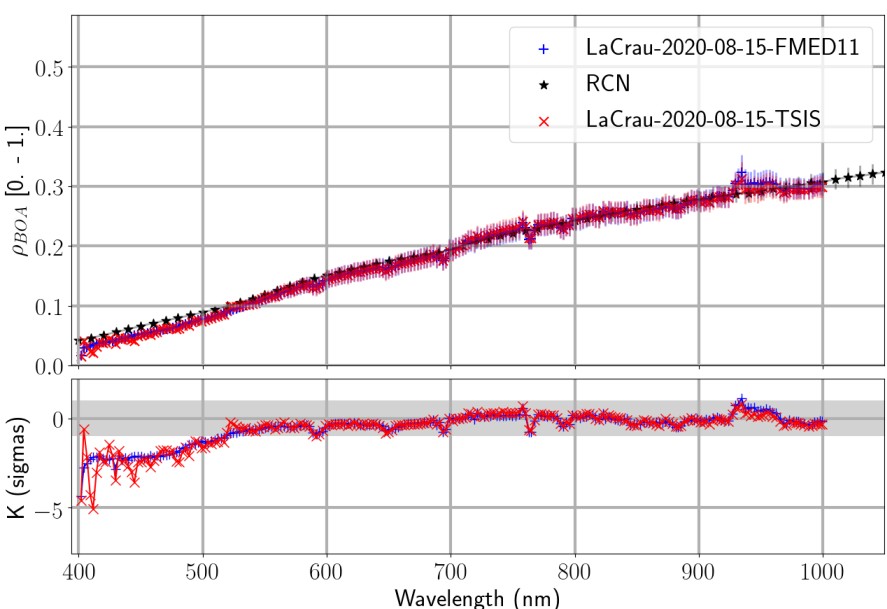

**Figure 12.** Inconsistent scenario for DESIS. Top: L2A surface reflectance of RCN (black crosses), PACO L2A with Fontenla 2011 (blue "+") and TSIS (red "X"s) solar models. Bottom: Uncertainty ratio (K) between each of L2A surface reflectance with the previous solar models with respect to RadCalNet (RCN) in situ reference values. Grey band limits ± 1 sigma region.

## 7. Conclusions

Recent updated measurements of the solar irradiance have shown differences at certain wavelengths with respect to current the solar irradiance spectra used by different missions. These differences are wavelength dependent and they are different for each of the solar spectra, as found in Thuillier 2003 and Fontenla 2011. The effect of these solar spectra differences is important in the atmospheric correction algorithms and, therefore, in the L2A products.

Our study concludes that when consistently using the same solar model to define the input radiance and for the atmospheric processor, then there are no significant differences in the L2A products.

When the atmospheric correction processor applies a different solar model (e.g., TSIS or Thuillier ) than the one used in the L1C radiance estimation (Fontenla), then some differences appear in the L2A ground reflectance in the SWIR bands, especially when using

the Thuillier 2003 model. These differences with TSIS are negligible when using Fontenla 2011 model.

Only differences in the solar model spectra of ∼10% in the blue wavelengths create some larger differences in the L2A BOA reflectance results. However, these differences are below the typical PACO BOA uncertainties at these wavelengths.

The differences in the estimation of the atmospheric parameters are, however, not significant, proving that the radiance discrepancies in the atmospheric simulations have the largest influence.

The mixing of solar irradiance models, between the TOA radiance estimate and the L2A model, is strongly discouraged by the authors, since it creates an inconsistency in the remote sensing processor chain.

The results also show that the difference between using Fontenla 2011 and TSIS 2021 in the L2A processor is negligible in the final products. Therefore, for other missions based on the Fontenla 2011 solar model (e.g., EnMAP [22]), an update of the solar model to TSIS 2021 irradiance spectrum does not seem to be necessary since it will not change the mission products significantly.

Further studies comparing with in situ measurements with different ground characteristic would add more statistics to these results.

**Author Contributions:** Conceptualization, R.D.L.R., K.A. and B.L.; methodology, R.D.L.R. and R.R.; software, R.D.L.R. and R.R.; validation, R.D.L.R. and M.B.; formal analysis, R.D.L.R. and R.R.; investigation, R.D.L.R. and R.R.; resources, M.B., B.P. and R.R.; writing—original draft preparation, R.D.L.R. and R.R.; writing—review and editing, R.D.L.R., R.R., M.B., B.P., K.A. and B.L.; supervision, R.D.L.R. and B.P.; project administration, P.R.; funding acquisition, P.R. All authors have read and agreed to the published version of the manuscript.

**Funding:** This research received partial funding by EU and ESA. The views expressed herein can in no way be taken to reflect the official opinion of the European Space Agency or the European Union.

**Data Availability Statement:** Not applicable

**Conflicts of Interest:** The authors declare no conflict of interest.

## Abbreviations

The following abbreviations are used in this manuscript:

| | |
|---|---|
| AC | Atmospheric Correction |
| AERONET | AErosol RObotic NETwork |
| AOT | Aerosol Optical Thickness |
| APDA | Atmospheric Precorrected Differential Absorption |
| BOA | Bottom-Of-Atmosphere |
| BRD | Bi-directional Reflection Distribution |
| CEOS | Committee on Earth Observation Satellites |
| DDV | Dark Dense Vegetation |
| DEM | Digital Elevation Model |
| DESIS | DLR Earth Sensing Imaging Spectrometers |
| EnMAP | Environmental Mapping and Analysis Program |
| ESA | European Space Agency |
| GSICS | Global Space-based Inter-Calibration System |
| IDL | Interactive Data Language |
| LUT | Look-Up-Table |
| MODTRAN | MODerate resolution atmospheric TRANsmission |
| NDVI | Normalized Difference Vegetation Index |
| RadCalNet / RCN | Radiometric Calibration Network |
| S2 | Sentinel-2 |
| SAA | Sun Azimutal Angle |
| SRF | Spectral Response Function |
| SRTM | Shuttle Radar Topography Mission |

| SZA | Sun Zenith Angle |
| TOA | Top-Of-Atmosphere |
| TSIS | Total and Spectral Solar Irradiance Sensor |
| VNIR | Visible and Near-InfraRed |
| SWIR | Short Wavelength InfraRed |
| WV | Water Vapor |

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
