# Peer review of "Influence of the Solar Spectra Models on PACO Atmospheric Correction"

_remotesensing, doi:10.3390/rs14174237_

Round 1
Reviewer 1 Report
This manuscript studies the influence of solar irradiance model selections on the radiance measurement and atmospheric correction algorithms. The authors suggest that a consistent solar irradiance model should be used in radiance measurement and down-streaming atmospheric correction algorithms. Some comments are as follows.
1. The authors mention L2A, L1C, etc. products for many times. More specific definitions should be given to the different processing levels.
2. Line 96, the full spelling of GSICS is not given.
3. Line 107, a redundant bracket after ‘FWHM=0.4 nm’.
4. Line 483, ‘understation’ may be a typo.
Author Response
Dear reviewer,
Thank you very much for your time to review the paper and for your comments.
A better introduction to the L1C and L2A products is now inside the Introduction.
Typos have been now corrected.
Changes will be found in orange (I have not managed to highlight the sections and abstract) and highlighted text in the new version of the article.
The previous section 2 is now split into several sections, reordering the contents for an easier and more intuitive understanding.
Best regards,
Reviewer 2 Report
Dear authors, many thanks for this interesting manuscript. You will find my comments attached. I recommend that you make use of a professional proofreader in the next iteration since the english needs to be improved and polished. Also the conclusion really needs to be entirely rewritten, as well as the start of the paper. The reason of this study is not clear enough and many abreviations are introduced without beeing explained. This makes the reading difficult. I would be please to review the improved version of your work.

Author Response
Dear reviewer,
Thank you very much for your time to review the paper and for all the comments.
We have tried to follow your recommendations detailed in all of them.
Changes will be found in orange (I have not managed to highlight the sections and abstract) and highlighted text in the new version of the article.
- The section 1.2 is now split in several sections, which are reordered following your detailed recommendations.
- In general, the abbreviations can be found at the end of the document. Before the bibliography. But we have also mentioned them in the text for completeness.
- We attach here also your PDF with your reviewed comments:
- Comments in green -> addressed in the text
- In red: answered comments
Best regards,

Round 2
Reviewer 2 Report
Dear authors, many thanks for having addressed my comments. There are still a few typos left which can be corrected before publishing. I appreciate your effort to reorganize the sections that were not clear and can accept the paper now.